# Increased Depressive-like, Anxiety-like, and Perseverative-like Behavior in Binge Eating Model in Juvenile Rats

**DOI:** 10.3390/nu16091275

**Published:** 2024-04-25

**Authors:** Alma Delia Genis-Mendoza, Isela Esther Juárez-Rojop, Yudy Merady Escobar-Chan, Carlos Alfonso Tovilla-Zárate, María Lilia López-Narváez, Humberto Nicolini, Thelma Beatriz González-Castro

**Affiliations:** 1Laboratorio de Genómica de Enfermedades Psiquiátricas y Neurodegenerativas, Instituto Nacional de Medicina Genómica, Ciudad de México 14610, Mexico; adgenis@inmegen.gob.mx; 2División Académica de Ciencias de la Salud, Universidad Juárez Autónoma de Tabasco, Villahermosa 86100, Mexico; ijr01127@docente.ujat.mx (I.E.J.-R.); 201e57001@egresados.ujat.mx (Y.M.E.-C.); 3División Académica Multidisciplinaria de Comalcalco, Universidad Juárez Autónoma de Tabasco, Comalcalco 86650, Mexico; mln07271@docente.ujat.mx; 4División Académica Multidisciplinaria de Jalpa de Méndez, Universidad Juárez Autónoma de Tabasco, Jalpa de Méndez 86205, Mexico; tgc05410@docente.ujat.mx

**Keywords:** binge-like eating, depressive-like, anxiety-like, perseverative-like, animal model

## Abstract

The aim of the present study was to evaluate depressive-like, anxiety-like, and perseverative-like behaviors in a binge eating model. Juvenile Wistar rats, using the binge eating model, were compared to caloric restriction, induced stress, and control groups. Rats of the induced stress group presented binge-like behaviors in standard food intake in the second cycle of the experiment when compared to the caloric restriction group and the binge eating model group. Depressive-like behavior was observed in the binge eating model group with longer immobility time (*p* < 0.001) and less swim time (*p* < 0.001) in comparison to the control group. Anxiety-like behavior was observed by shorter duration of burying latency in the binge eating model group when compared to the induced stress group (*p* = 0.04) and a longer duration of burying time when compared to the control group (*p* = 0.02). We observed perseverative-like behavior by the binge model group, who made more entries to the new arm (*p* = 0.0004) and spent a longer time in the new arm when compared to the control group (*p* = 0.0001). Our results show differences in behaviors between the groups of rats studied. These results suggest that calorie restriction–refeeding, along with stress, may lead to depressive-like, anxiety-like, and perseverative-like behavioral changes in male Wistar rats.

## 1. Introduction

Binge eating disorder (BED) is an eating behavior disorder characterized by excessive food intake in a short time (binge eating), with a loss of control over that intake [1]. The literature indicates that binge eating is characterized by repeated, intermittent, and exaggerated food intake in short periods of time (within a time range of 2 h) [1]; the minimum incidence of binge eating is once a week over three months. According to the Diagnostic and Statistical Manual of Mental Disorders (DSM-5) criteria, the amount of food eaten is greater than what most people would consume under similar circumstances and in a similar period of time. Binge eating may manifest itself even in the absence of hunger with feelings of shame, depression, guilt, and loss of control [2]. In addition, binge eating is associated with behavioral and affective alterations, as well as obesity [3,4]. It is suggested that self-imposed dieting, coupled with overeating, precedes binge eating behaviors in humans [2,4].

Binge eating disorder is the most common eating disorder [4]. Studies in children with obesity indicate the prevalence of binge eating disorder [2,5,6]. Studies in the adolescent population indicate a high prevalence of adolescents developing this disorder (7.8%) and its association with obesity. Research indicates a higher prevalence of binge eating disorder in the male population (11.1%) than in than female population (4.4%); a higher presence of obesity in adolescents with binge eating disorder (33.3%) than in adolescents without binge eating disorder (10.4%) has also been observed [2]. However, there are reports of higher incidences of binge eating disorder in female adolescents (75.10%) than in male adolescents (24.90%), with anxiety, depression, and social phobia [6].

Thus, binge eating disorder affects both male and female adolescents. Also, binge eating disorder interferes with decision making [7]. Individuals with binge eating disorder and obesity make decisions that involve risk, with high probabilities of loss and moderate probabilities of reward. This manifests a deficiency in decision making [7]. Therefore, models are necessary to explain this behavior in binge eating.

To improve our understanding of animal models for binge eating, a proper characterization of animal models is necessary. Smith suggests a classification of four types of animal models of eating disorders: etiological, isomorphic, mechanistic, and predictive [8]. Isomorphic models are designed to resemble symptomatology in humans (similarity in behavioral states) [9]. Up to now, all the animal models in binge-like eating are type isomorphic models. There are different animal models of binge eating, with their different points of validity. The literature indicates that animal models are characterized by diet and refeeding cycles, caloric restriction–refeeding and/or induced stress cycles, simulated refeeding, behavioral pattern refeeding, and limited access to optional foods [10]. It should be noted that no animal model exactly mimics human binge eating; nonetheless, animal models are considered useful, since limited access to palatable food at a given time promotes repeated episodes of binge-like eating [11].

Some studies using animal models have suggesting that caloric restriction and stress promote binge-like eating [12] and caloric restriction increases sensitivity to hyperphagic effect [13] and increased expression of sterol regulatory element binding protein in the hypothalamus [14]; at the same time, binge eating produces dopamine and μ-opioid receptor dysregulation in the brain [15]. Moreover, access to highly palatable foods (HPFs) increases anxiety-like behavior [16,17] and food addiction [18].

There is limited evidence in animal models of binge-like eating that simultaneously assesses depressive-like, anxiety-like, and perseverative-like behaviors [19]. Therefore, the aim of this study was to assess depressive-like, anxiety-like, and perseverative-like behaviors in a juvenile Wistar rats in binge eating, caloric restriction, induced stress, and control groups. We used the forced swimming test to assess depressive-like behaviors [20], the defensive burying test to assess anxiety-like behaviors [21], and the Y-maze test to assess perseverative-like behaviors [22]. The animal model used in the present study is an adaptation of characteristics of the Boggiano and the Cifani model [23,24], but we used a combined of stress + calorie restriction and a combination of compulsion in eating and weight. We considered this as and advancement over the previous models.

## 2. Materials and Methods

### 2.1. Animals

The experiment was performed using a total of 32 male Wistar rats of thirty days old, obtained from the animal colony of the Unidad de Producción, Cuidado y Experimentación Animal of the División Académica de Ciencias de la Salud (DACS), Universidad Juárez Autónoma de Tabasco (UJAT). The number of male rats included in this study was determined based on the suggestions of the Boggiano model, with a minimum of 24 animals [23]. Rats were randomly divided into four groups and were kept in acrylic cages. Each group of rats in a cage (*n* = 8 per cage) were under light/dark cycles of 12 h (light at 08:00 h), constant humidity, and 20–22 °C temperature [24], with the minimum spaces according to the recommendations of Official Mexican Standard NOM-062-ZOO-1999 technical specifications for the production, care, and use of laboratory animals [25]. Bioethics was considered in the use of experimental animals [26], as well as criteria for the handling and use of experimental animals by the internal committee of UJAT-DACS (Comité Interno para el Cuidado y Uso de los Animales de Laboratorio-CICUAL).

### 2.2. Experimental Design

The binge eating model used in this study was an adaptation of the Boggiano model and the Cifani model [23,24]. Our binge eating model combines three cycles of caloric restriction–refeeding plus repeated stress induction to examine depressive-like, anxiety-like, and perseverative-like behaviors in Wistar rats. The experimental design is shown in Figure 1.

### 2.3. Diets

Rats were fed two types of diets: standard diet (5001 LabDiet; 2.8 kcal/g) for maintenance in pellet form [27] and a highly palatable food [24].

The rats underwent an initial adaptation period of three days with free access to food and water. After the adaptation period, each group received a corresponding diet with caloric restriction–refeeding cycles. Highly palatable food was a textured paste prepared with Nutella (Ferrero, Alba, Italy) chocolate cream (5.55 kcal/g; carbohydrates 55.5%, fat 33.3%, protein 5.5%), ground pellets (LabDiet; 2.8 kcal/g), and water. The proportions were: 52% Nutella, 33% ground pellets, and 15% water [24]. Rats did not have access to the palatable feed until day 5 of the first cycle of caloric restriction–refeeding. Palatable food was offered ad libitum for 2 h during the light cycle. Stress-free groups (control and caloric restriction) received cups with palatable food between 10:00 and 12:00 h. Stress groups (induced stress and binge eating model) received the palatable food between 10:15 and 12:15 h.

### 2.4. Caloric Restriction–Refeeding Cycles Plus Stress

According to the suggestions of the Boggiano [23] and Cifani models [24], we paired the groups of rats into weight ranges. The groups were classified as shown in Table 1.

#### 2.4.1. Group One: Control (Restriction—No, Stress—No)

The control group received the following in cycle 1: standard food ad libitum during the first 4 days; on days 5–6 they received standard food ad libitum, plus palatable food (for 2 h); on days 7–8 they received standard food ad libitum. During cycle 2, the control group received standard food ad libitum on days 9–12; on days 13–14 they received standard food ad libitum, plus palatable food for a 2 h period; on days 15–16 standard food ad libitum was provided. In cycle 3, the control group was given standard food ad libitum on days 17–24; on day 25 they were given standard food ad libitum, plus palatable food for a period of 2 h. This group was not exposed to stress during the whole experiment.

#### 2.4.2. Group Two: Caloric Restriction (Restriction—Yes, Stress—No)

During cycle 1, the caloric restriction group received standard food restriction (restriction-1) on days 1–4; on days 5–6 standard food was offered ad libitum, plus palatable food (for 2 h); on days 7–8 only standard food ad libitum was given. During cycle 2, this group was restricted in its standard food (restriction-2) on days 9–12; on days 13–14 standard food was provided ad libitum, plus palatable food for a period of 2 h; on days 15–16 only standard food was provided ad libitum. During cycle 3, the caloric restriction group received standard food restriction (restriction-3) on days 17–20; on days 21–24 they received standard food ad libitum; on day 25 they received standard food ad libitum, plus palatable food for a 2 h period.

#### 2.4.3. Group Three: Induced Stress (Restriction—No, Stress—Yes)

During cycle 1, the induced stress group received standard food ad libitum on days 1–4; on days 5–6 they received standard food ad libitum, plus palatable food (for 2 h) to induce dysfunctional eating behavior; on days 7 and 8, the group was exposed to a stress procedure and received standard food ad libitum. During cycle 2, on days 9–12 this group received standard food ad libitum; on days 13–14 they received standard food ad libitum, plus palatable food for a 2 h period, and were exposed to stress; on days 15–16 they were given standard food ad libitum. During cycle 3, the induced stress group received standard food ad libitum on days 17–24; on day 25 they received standard food ad libitum, plus palatable food for a 2 h period, and were exposed to stress.

#### 2.4.4. Group Four: Binge Eating Model (Restriction—Yes, Stress—Yes)

During cycle 1, the binge eating model group received standard food restriction (restriction-1) on days 1–4; on days 5–6 they received standard food ad libitum, plus palatable food (for 2 h), and to induce dysfunctional eating behavior they were exposed to a stress procedure; on days 7–8 they received standard food ad libitum. During cycle 2, this group was restricted in its standard food (restriction-2) on days 9–12; on days 13–14 they received standard food ad libitum, plus palatable food for a period of 2 h, and were exposed to stress; on days 15–16 they received standard food ad libitum. During cycle 3, the binge eating model group received standard food restriction (restriction-3) on days 17–20; on days 21–24 they received standard food ad libitum; on day 25 they received standard food ad libitum, plus palatable food for a period of 2 h, and were exposed to stress.

### 2.5. Standard Food Restrictions

The food restriction (in percentage) during each cycle followed the restrictions of the Cifani model [24]. Restriction-1: rats received 66%, equivalent to 2/3 of the entire group standard food intake of the 3 days prior to the beginning of the experiment, corresponding to the adaptation period. The group had food restricted for 4 days. Restriction-2 and restriction-3: rats received 66%, equivalent to 2/3 of the standard food intake of nonrestricted rats during the last 2 days of the cycles where rats received only standard food ad libitum.

### 2.6. Stress-Induction Procedure

The induced stress and binge eating model groups received stress induction in each cycle of caloric restriction–refeeding. Over a period of 15 min, palatable food was placed in a coffee cup; this container was placed outside the cages, on top of the grid lids where the pelleted food is normally placed. The rats were able to observe and smell the cup containing palatable food but did not have access to it. The rats were subjected to this stress condition from 10:00 h to 10:15 h. After 15 min, the cup with palatable food was placed inside the rats’ cages, and the amount of palatable food consumed over a period of two hours (10:15 h–12:15 h) was quantified. Induced stress was performed on days 5, 6, 13, 14, and 25 of the experiment [24] (Table 1).

### 2.7. Behavioral Tests

We applied three behavioral tests at the end of the three cycles of caloric restriction–refeeding (one behavioral test per day), in the following order: forced swimming test to examine depressive-like behaviors; defensive burying test to examine anxiety-like behaviors; and Y-maze test to examine perseverative-like behaviors. Rats were tested individually. The tests were performed at 09:00 h and recorded with the aid of a video camera mounted on a tripod (Sony Handycam Hybrid, Dcr-sr45, manufacturer in China), for subsequent behavioral readout. Researchers who evaluated behavioral metrics (with respect to the four groups of rats) were blinded. Three researchers evaluated each test.

### 2.8. Forced Swim Test (Depressive-like)

The forced swimming test was used to assess depressive-like behavior. We used two rooms, a waiting room and an experimental room. The experiment consisted of two sessions separated by an interval of 24 h [20]. In the first session, the rats were induced to swim in a forced manner (15 min of training). We placed each rat into a vertical cylinder container (50 cm in height and 20 cm in diameter) filled with water (23 °C) to a height of 30 cm, according to Lucki’s modifications [28] as described in other studies [29,30], so that the rat could float and swim without its paws or tail touching the bottom of the cylinder. The container was cleaned, and the water was changed after each session. At the end of the training session, each rat was dried and placed in a dry box with a lamp to provide heat. The second session lasted 5 min, and the forced swim was recorded with a video camera. Immobility, swimming, and climbing times were quantified. Depressive-like behavior was related to longer immobility time and less time spent swimming and climbing [31,32].

### 2.9. Defensive Burying Test (Anxiety-like)

The defensive burying test was used to evaluate anxiety-like behavior in rats [21]. Defensive burying is described as the rat moving or displacing bedding material (classic shavings made of natural wood) with forward pushing movements (with the front paws) and shoveling movements (with the snout) directed toward the threat source [33]. De Boer and Koolhaas conducted a review of defensive burial and suggest that decreased burial latency time, increased burial time, and increased immobility time refer to anxiety-like behavior [34]. We used a polyethylene chamber with a lid (40 cm long, 30 cm wide, and 40 cm high). The introduction of an electrode through a hole in a wall of the box (located 2 cm above a 5 cm-thick bed of shavings) facilitated 3 mA discharges. Each rat was introduced into the box, in a corner opposite the electrode. The lid of the box was then placed to prevent the rat from escaping after the electric shock. The execution of this test lasted 5 min, during which we recorded the number of scans to the electrode, latency of burying behavior, burying time, immobility time, number of electric shocks, and reactivity to electric shock. Between each test, we replaced the bedding shavings. This test was recorded on a video camera (Sony Handycam Hybrid, Dcr-sr45, manufacturer in China) with a red light. From observation of the recordings, the behavioral results were recorded manually.

### 2.10. Y-Maze Test (Perseverative-like Behavior)

The Y-maze test was performed to assess perseverative-like behavior [35,36], since perseveration has been analyzed in animal models of compulsive behavior [37]. We used a Y-shaped maze; the three arms were 50 cm long and 10 cm wide, oriented at a 120° angle to each other. Each arm was structured with guillotine-shaped doors separating the main arm from the two arms of the maze body. The Y-maze was divided into 4 zones to facilitate the reading of the rats’ behavior: entry zone (entry arm of the rat at the beginning of the sessions); novelty zone (blocked arm during the training session/new arm during the test session); known zone (open arm in the training session/known arm in the test session); and middle zone of the maze (area where the entry zone, novel zone, and known zone converge). This test consisted of two sessions: training session (the entrance of one arm of the maze was blocked, and the rat was allowed to explore the maze for 5 min) and the test session (the arms of the maze unblocked; the rat explored the maze for 5 min), with a time interval of one hour between sessions. Increased inputs to an arm are an indicator of compulsive behavior [38]. We considered an arm entry to be when the rat returned to the same distal arm after visiting the central area of the Y-maze (the rat chose an arm when it placed all 4 paws on the arm). We quantified the exploration time in each arm of the maze.

### 2.11. Statistical Analysis

To determine the normality of the data, we performed a Shapiro–Wilk test. The variables had a normal distribution (*p*-value < 0.05). We used an ANOVA test to assess the effect of caloric restriction–refeeding cycles between groups. We used one-way ANOVA to assess the effect of caloric restriction–refeeding cycles on the tests that we used for measuring depressive-like, anxiety-like, and perseverative-like behaviors. Tukey’s post hoc test was used to compare the mean difference values of the groups. The level of statistical significance was set at *p* < 0.05. Data are presented as means ± standard error of mean. The data were analyzed by GraphPad Prism 6.1 software (GraphPad Software, Boston, MA, USA).

## 3. Results

### 3.1. Effect of Caloric Restriction–Refeeding Cycles Plus Stress on Binge Eating

#### 3.1.1. Pellet Intake

Pellet intake during the 25 days of the experiment by group was the following: control group (20.84 ± 0.89 g), caloric restriction group (17.20 ± 1.29 g), induced stress group (20.39 ± 0.75 g), and binge eating model (17.03 ± 1.24 g). The results of the overall ANOVA statistical analysis with Tukey’s post hoc test revealed differences in pellet intake between the four groups [F (3,96) = 3.591, *p* = 0.0165]. When analyzing the intake for each caloric restriction–refeeding cycle, we observed statistically significant results only in the second cycle (Figure 2A). Cycle one resulted in control group (17.31 ± 0.67 g), caloric restriction group (13.89 ± 1.72 g), induced stress group (17.46 ± 0.70 g), and binge eating model group (13.97 ± 1.77 g). No statistically significant differences were observed between groups [F (3,28) =2.249, *p* = 0.1046]. Cycle two resulted in control group (20.22 ± 1.28 g), caloric restriction group (15.43 ± 1.09 g), induced stress group (20.05 ± 1.18 g), and binge eating model (15.31 ± 1.09 g). We observed statistically significant differences between groups [F (3,28) =5.597, *p* = 0.0039]. Post-test analyses showed differences of means between the caloric restriction group and the control group (mean diff. −4.79, *p* = 0.0335); between the binge eating model group and the control group (mean diff. −4.91, *p* = 0.0281); between the induced stress group and the caloric restriction group (mean diff. 4.62, *p* = 0.0422); and between the binge eating model group and the induced stress group (mean diff. −4.74, *p* = 0.0356). We did not observe any differences of means between the induced stress group and the control group (mean diff. −0.16, *p* = 0.9996), or between the binge eating model group and the caloric restriction group (mean diff. −0.12, *p* = 0.9998). Cycle three resulted in control group (24.54 ± 1.40 g), caloric restriction group (21.73 ± 2.56 g), induced stress group (23.28 ± 1.09 g), and binge eating model group (21.27 ± 2.42 g). ANOVA did not reveal statistical significance [F (3,32) = 0.5728, *p* = 0.6370].

#### 3.1.2. Highly Palatable Food Intake

Highly palatable food intake by groups included the following: control group (6.82 ± 1.67 g), caloric restriction group (9.91 ± 1.80 g), induced stress group (6.28 ± 1.39 g), and binge eating model group (8.26 ± 1.98 g). ANOVA statistical analysis (one-way) with Tukey’s post hoc test did not reveal statistically significant results [F (3,16) = 0.8839, *p* = 0.4704]. Thus, we did not observe any differences of means between the caloric restriction group and the control group (mean diff 3.08, *p* = 0.5976), between the induced stress group and the control group (mean diff −0.53, *p* = 0.9961), the binge eating model group and the control group (mean diff 1.44, *p* = 0.9335), the induced stress group and the caloric restriction group (mean diff −3.62, *p* = 0.4696), the binge eating model group and the caloric restriction group (mean diff −1.64, *p* = 0.9057), and the binge eating model group and the induced stress group (mean diff 1.97, *p* = 0.8488) (Figure 2B).

#### 3.1.3. Body Weight

Body weight by group included control group (203.0 ± 9.24 g), caloric restriction group (187.4 ± 9.43 g), induced stress group (207.9 ± 9.59 g), and binge eating model group (217.3 ± 13.92 g). The overall ANOVA statistical analysis with Tukey’s post hoc test gave not statistically significant differences among the four groups when body weight data in the three caloric restriction–refeeding cycles were considered together [F (3,96) = 1.351, *p* = 0.2624]. Comparison of body weight by each caloric restriction–refeeding cycle is shown in Figure 2C. Cycle one resulted in control group (148.6 ± 7.09 g), caloric restriction group (136.5 ± 6.51 g), induced stress group (150.6 ± 6.88 g), and binge eating model group (141.1 ± 6.99 g). The overall ANOVA with Tukey’s post hoc test did not reveal statistically significant results [F (3,28) = 0.9169, *p* = 0.4454]. Cycle two resulted in control group (202.7 ± 5.40 g), caloric restriction group (184.9 ± 9.60 g), induced stress group (208.4 ± 6.18 g), and binge eating model group (214.7 ± 17.60 g). ANOVA statistical analysis with Tukey’s post hoc test did not reveal statistically significant results between groups [F (3,28) =1.400, *p* = 0.2635]. Cycle three resulted in control group (248.8 ± 4.60 g), caloric restriction group (239.5 ± 7.58 g), induced stress group (260.9 ± 3.83 g), and binge eating model (292.4 ± 8.17 g). The ANOVA analysis with Tukey’s post hoc test did reveal statistically significant differences [F (3,28) =13.30, *p* < 0.0001]. Post-test analysis showed differences of means between the binge eating model group and the control group (mean diff. 43.64, *p* = 0.0002), between the binge eating model group and the caloric restriction group (mean diff. 52.94, *p* < 0.0001), and between the binge eating model group and the induced stress group (mean diff. 31.55, *p* = 0.0076).

### 3.2. Effect of Caloric Restriction–Refeeding Cycles on Forced Swim Test Behavior

The immobility times by group were control group (10.60 ± 1.67 s), caloric restriction group (11.20 ± 3.49 s), induced stress group (26.60 ± 4.15 s), and binge eating model (28.60 ± 4.27 s). Statistically significant differences were observed between groups [F (3,12) = 42.94, *p* < 0.001]. Post-test analysis showed differences of means between the control group and induced stress group (mean diff. 16.00, *p* < 0.001) and binge eating model group (mean diff. 17.40, *p* < 0.001). Also, differences were observed between the induced stress group and the caloric restriction group (mean diff. 15.40, *p* < 0.001) and between the binge eating model group and the caloric restriction group (mean diff. 18.80, *p* < 0.001) (Figure 3A).

The times spent swimming by group were control group (133.8 ± 15.99 s), caloric restriction group (75.80 ± 22.26 s), induced stress group (55.86 ± 15.23 s), and binge eating model (93.40 ± 8.62 s). Statistically significant differences were observed between the groups [F (3,12) = 20.94, *p* < 0.001]. Post-test analysis showed differences of means between the control group and caloric restriction group (mean diffs −58.00, *p* < 0.001), induced stress group (mean diff. −76.80, *p* < 0.0001), and binge eating model group (mean diff. −40.40, *p* < 0.05), and between the binge eating model group and the induced stress group (mean diff. 36.40, *p* < 0.05) (Figure 3B).

The times spent climbing by group were control group (155.0 ± 17.56 s), caloric restriction group (197.6 ± 23.33 s), induced stress group (216.3 ± 21.79 s), and binge eating model (171.2 ± 15.53 s). Statistically significant differences were observed between the groups [F (3,12) = 12.12, *p* < 0.001]. Post-test analysis showed differences of means between the control group and caloric restriction group (mean diff. −42.60, *p* < 0.05) and induced stress group (mean diff −61.40, *p* < 0.001). Also, differences were observed between the binge eating model group and induced stress group (mean diff. 45.20, *p* < 0.01) (Figure 3C).

### 3.3. Effect of Caloric Restriction–Refeeding Cycles on Defensive Burying Test

When we observed the latency of burying behavior, statistical differences were observed between groups [F (3,18) =3.23, *p* = 0.04]. The control group (34.86 ± 12.68 s) and binge eating model group showed the lower time of latency (34.33 ± 11.64 s) in comparison to the caloric restriction group (38.00 ± 12.70 s) and induced stress group (52.57 ± 8.05 s). However, post-test analysis showed no differences of means between groups (Figure 4A).

Also, statistical differences were observed between groups when we analyzed the burying time [F (3,15) =4.09, *p* = 0.02]. The control (137.8 ± 23.16 s) and caloric restriction groups (137.6 ± 18.61 s) show a lower burying time. Contrarily, the induced stress (149.7 ± 20.40 s) and binge eating model (174.7 ± 15.25 s) groups showed the highest time of burying. Post-test analysis showed differences of means between the control group and binge eating model group (mean diff. 36.83, *p* < 0.05) and between the binge eating model group and the caloric restriction group (mean diff. 37.07, *p* = 0.0307) (Figure 4B).

Finally, we evaluated the mobility time by group. The results were not statistically significant among the four groups [F (3,19) = 1.520, *p* = 0.2415]; results included control group (13.67 ± 11.62 s), caloric restriction group (32.00 ± 21.77 s), induced stress group (18.33 ± 8.80 s), and binge eating model group (20.67 ± 14.68 s).

In the defensive burying test, we observed the number of scans to the electrode by group: control group (8.25 ± 5.03), caloric restriction group (9.25 ± 4.33), induced stress group (7.50 ± 3.74), and binge eating model (9.00 ± 5.12). The results were not statistically significant among the four groups [F (3,28) = 0.2369, *p* = 0.8699]. The number of electric shocks by group included control group (1.16 ± 0.40), caloric restriction group (1.40 ± 0.54), induced stress group (1.83 ± 0.40), and binge eating model group (1.83 ± 0.40). Statistically significant differences were observed between the four groups [F (3,19) = 3.342, *p* = 0.0411]. The post-test analysis no showed difference of mean. Reactivity to electric shock by group included control group (2.16 ± 0.40), caloric restriction group (2.60 ± 0.54), induced stress group (3.33 ± 0.81), and binge eating model group (3.16 ± 0.98). Statistically significant differences were observed between the four groups [F (3,19) = 3.169, *p* = 0.0481]. Post-test analysis showed no differences of means between the induced stress group and the control group (mean diff. 1.16, *p* = 0.05).

### 3.4. Effect of Caloric Restriction–Refeeding Cycles on Y-Maze Behavior

Statistically significant differences were observed when we compared the novel arm by group [F (3,21) = 6.24, *p* = 0.003]. The percentages by groups were control (51.43 ± 10.34%), caloric restriction (49.52 ± 13.11%), induced stress (41.67 ± 9.63%), and binge eating model (79.17 ± 22.44%). Likewise, in comparison with the control group, we observed that only the binge eating model group showed statistical differences (mean diff. −27.74, *p* = 0.01). But differences were observed between the binge eating model group and the caloric restriction group (mean diff. −29.64, *p* = 0.01), as well as between the binge eating model group and the induced stress group (mean diff. −37.50, *p* < 0.01) (Figure 5A).

Regarding the time spent (in percentage) in the novel arm by group, differences were observed between groups [F (3,18) = 15.41, *p* < 0.0001]. Post-test analysis showed differences of means between the control group when compared with the induced stress group (mean diff. 24.24, *p* < 0.05). Likewise, there were differences of means with the caloric restriction group (mean diff. 32.15, *p* < 0.05) and induced stress group (mean diff. 52.67, *p* < 0.00) in comparison with the binge eating model group. No differences of mean were observed between the caloric restriction or induced stress group and the control group (Figure 5B).

## 4. Discussion

The aim of the present study was to evaluate anxiety-like, depression-like, and perseverative-like behaviors in a juvenile binge eating model. There are few studies that have evaluated the presence of behavioral alterations such as depression, anxiety, and compulsion in the same group of rats in a binge eating model [39,40]. Our study showed that calorie restriction–refeeding, along with stress, may lead to anxiety-like, depression-like, and perseverative-like behaviors in rats in a binge eating model.

First, in this binge eating model, we combined two factors: caloric restriction and stress, in order to induce binge-like eating [1]. Cycles of caloric restriction–refeeding plus induced stress have been reported in previous studies of animal models of binge-like eating [23,24]. Animal models of binge-like eating with stress induction have led to research related to changes in body weight, obesity [41], behavioral changes [42], corticostriatal-hypothalamic circuitry, and food motivation [43]. We used 30-day-old adolescent male rats [44] to analyze the sensitivity to palatable foods in young rats, because studies of animal models of binge-like eating with adolescent rats are limited [45,46].

In contrast to the models of Boggiano and Chandler [23] and Cifani et al. [24], who included female Sprague-Dawley rats in their studies, we included Wistar rats to analyze whether behavioral sensitivity was maintained in binge-like eating when the strain and sex of the rats were switched. Some reports indicate that stress reduces the intake of highly palatable food in adult rats (male and female) [47]. In addition, studies in adult male rats support the involvement of the brain noradrenergic system [48] and limited access to highly palatable food in binge-like eating [49]. In this work, we only evaluated male Wistar rats, in part to explore the behavior without other variables; however, it is necessary to evaluate these behaviors in female Wistar rats, since, as is known, eating disorders occur more frequently in women. However, binge eating disorder is more common in men [50].

As suggested by previous studies, we applied a minimum of three cycles of caloric restriction plus stress [12,23]. In this experiment, we applied a type of mild stress to the rats that is ideal for inducing strong behavioral activation, but at the same time inhibits freezing and fear behaviors. This type of stress was based on the lack of control (for a short period of time) of environmental circumstances. In contrast, the induction of aggressive stress (such as foot shock) induces freezing and fear behaviors in rats [51]. The type of stress induction we used consisted of palatable food on top of the cage lid grids, as the rats were able to smell the palatable food and visualize the cup containing it. We considered that this type of stress is similar to that of those on restrictive diets, where people who want to lose weight restrict palatable foods, with the desire to lose weight, but this causes them stress. This provokes the behavior that when they finish the diet, they compulsively eat these palatable foods again.

Regarding the standard food intake in the second cycle, we observed that restriction alone or the combination of restriction plus stress, did not induce binge-like eating. Nonetheless, animals that only received stress induction (with free access to standard food) manifested binge-like eating. In contrast, highly palatable food intake in rats did not signal binge-like eating. It might be that more cycles of caloric restriction–refeeding favors binge-like eating with highly palatable food intake, as indicated by the findings of Hagan and Moss, where rats manifested binge-like eating with up to 12 weeks of refeeding–restriction needed [52]. Therefore, the stress factor coupled with repeated restrictions of food intake is important for the early onset of binge-like eating [12].

The study of depressive-like behavior in rats, with the forced swim test, has been reported since 1978. In this test, a longer time of passive behavior (immobility) and a shorter time of active behavior (swimming and climbing) are characteristic of depressive-like behavior [20,31]. Slattery and Cryan have pointed out the advantages and disadvantages of this preclinical model in mood disorders [29], although it has also been reported as a suitable test for assessing depression in rats [53]. Our results suggest the presence of depressive-like behavior based on longer immobility time and shorter time spent swimming in rats with only stress induction and in rats in the binge eating model group, compared to the control group. These results suggest that the combination of dietary restriction plus stress can lead to depressive-like behavioral alterations. As we have already pointed out, eating disorders are complex with several associated comorbidities, such as depression [1].

Evidence suggests the use of the defensive burying test to measure anxiety-like behavior in rats, via shorter latency time to burying, longer burying time, and longer immobility time [34]. In our study, the defensive burying test revealed that the combination of food intake restriction and stress induction precedes anxiety-like behavioral alteration, with longer burying time compared to only the restriction factor. On the other hand, Colautouni et al. provided a highly palatable type of food (sucrose), different from this study, to male Sprague-Dawley rats and observed anxiety-like behavior [54]. The aforementioned study suggests that binge-like eating coincides with anxiety-like behavior in rats, maintaining behavioral sensitivity. It has been suggested that the binge eating that people engage in is due to anxiety. However, as we have discussed, anxiety is only one of the comorbid disorders, which must be treated early.

Finally, The Y-maze test was applied to assess perseverative-like behavior in rats [36]. Restraint plus repeated induction of stress induced a preference for the unknown arm, with a higher number of entries and a longer dwell time. These results are similar to those found by Van de Vondervoort et al., who also found a strong increase in entries and time spent in the new arm [38]. Contrary to our results, Hussain and Krishnamurthy did not find perseverative-like behavior in rats with similar features of binge-like eating [19]. One of the characteristics of eating disorders are observations and compulsions, which translate into binge eating or compensatory behaviors, such as excessive exercise or vomiting, behaviors that we find exclusively in humans. In animal models, this type of evaluation is complicated; however with the Y-maze, we tried to evaluate these obsessive behaviors, which did not have statistical significance, but we want to continue trying this behavior with some modifications to see if its evaluation is possible.

Finally, we recognize some limitations in our study. We used only male rats. Eating disorders occur more often in women than in men, although recently an increase has been observed in men. Working with male rats facilitated behavioral readings, without taking into account hormonal variations. However, the next challenge is also working with female rats. Secondly, we measured in two phases, 30 or 90 days. Eating disorders in humans occur at the beginning of adolescence; however, it is increasingly occurring at younger ages. Studying 30-day-old animals helped us see similarities in adolescence. Our group has studied ages of 45 and 90 days, with similarities in adolescent and adult behavior, where it is possible to observe characteristics associated with psychiatric disorders, especially with schizophrenia. However, we consider as a strength of the present study that our animal model is an adaptation of characteristics of the Boggiano model and the Cifani model [23,24]. The cycles are combined (stress + calorie restriction) highlighting that with this combination, compulsion in eating and weight (binge eating) is observed. In addition, results of five tests are measured to determine behavioral hopelessness, anxiety, and compulsion. Such tests had not been carried out on a similar model. Our suggestions can be improved by increasing the number of cycles to work on a cleaner model of binge eating disorder.

## 5. Conclusions

In the present study, we combined caloric restriction–refeeding cycles plus repeated stress induction. Repeated stress induction produced binge-like eating when we measured standard food intake. Binge-like eating was not observed when ingesting highly palatable food. However, it was very noticeable that the combination of caloric restriction plus repeated stress induction preceded an increase in body weight. We found that juvenile rats subjected to repeated stress induction exhibited binge eating-like behavior and showed depression-like behavior, while rats with binge-like eating manifested depression-like, anxiety-like, and perseverative-like behaviors. These results suggest that caloric restriction–refeeding, coupled with repeated induction of stress, may be useful in the analysis of behavioral alterations such as depressive-like, anxiety-like, and perseverative-like behaviors in juvenile rats with binge-like eating. Nonetheless, it is still necessary to understand the biological mechanisms involved in the presence of depressive-like, anxiety-like, and perseverative-like behaviors in rats with binge-like eating and stress.

## Figures and Tables

**Figure 1 nutrients-16-01275-f001:**
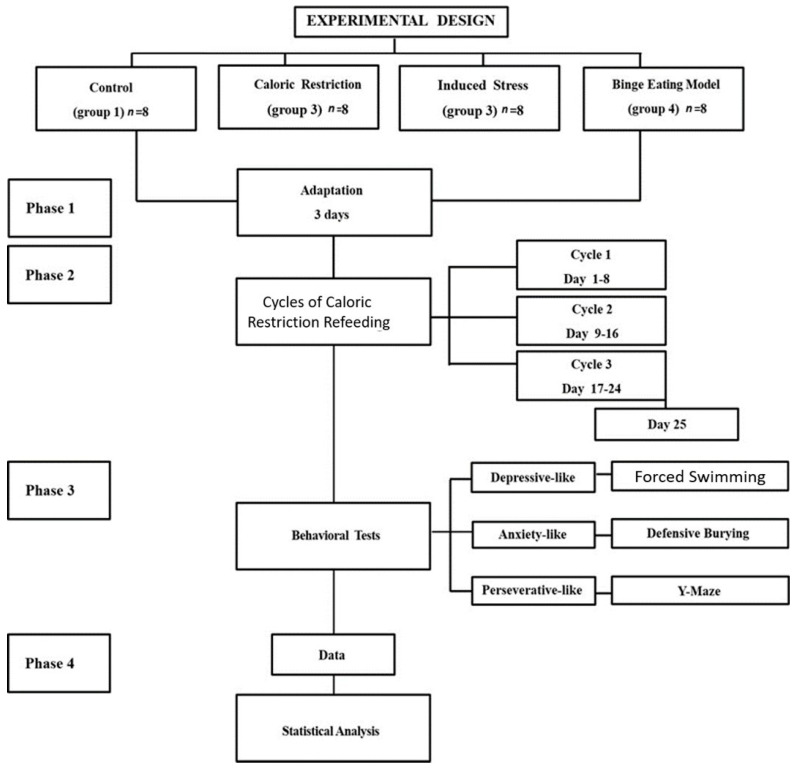
Flow diagram showing the different phases of the experiment. During the second phase, the restriction was of standard food (pellets), and the refeeding was with HPF (Highly Palatable Food).

**Figure 2 nutrients-16-01275-f002:**
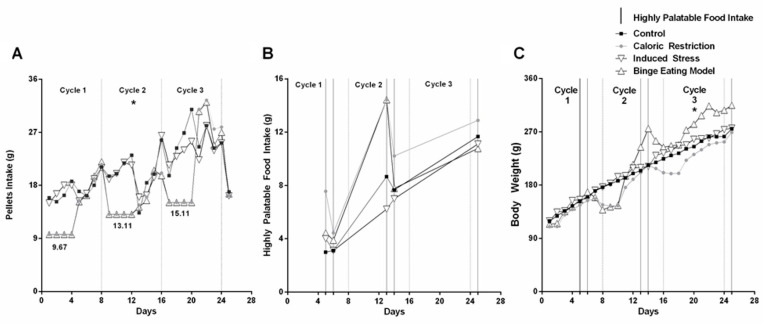
(**A**) Pellet Intake. Variations of standard food intake (in grams) of the four groups of rats during the three cycles of the experiment. Binge eating occurred in the group of stressed rats in the second cycle. Binge-like eating occurred in the induced stress group in the second cycle. The graph shows the grams of pellets provided to the caloric restriction groups (caloric restriction group and binge eating model group) during each cycle. (**B**) Highly palatable food intake (in grams) of the four groups of rats, during the three cycles of caloric restriction–refeeding. (**C**) Body weight variations (in grams) of the four groups of rats, during the three cycles of caloric restriction–refeeding. * *p* < 0.05.

**Figure 3 nutrients-16-01275-f003:**
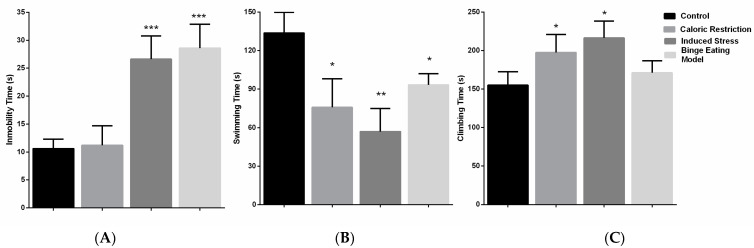
Depressive-like behavior. (**A**) Immobility time (in seconds) of the four groups of rats. (**B**) Swimming time (in seconds) of the four groups of rats. (**C**) Climbing time (in seconds) of the four groups of rats. * *p* < 0.05, ** *p* < 0.01, *** *p* < 0.001.

**Figure 4 nutrients-16-01275-f004:**
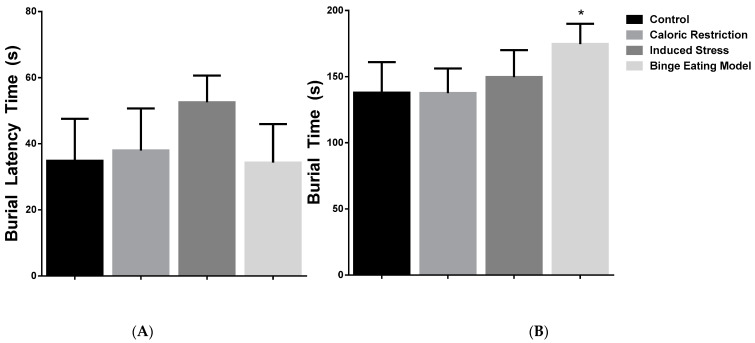
Anxiety-like behavior. (**A**) Burial latency time (in seconds) of the four groups of rats. (**B**). Burial time (in seconds) of the four groups of rats. * *p* < 0.05.

**Figure 5 nutrients-16-01275-f005:**
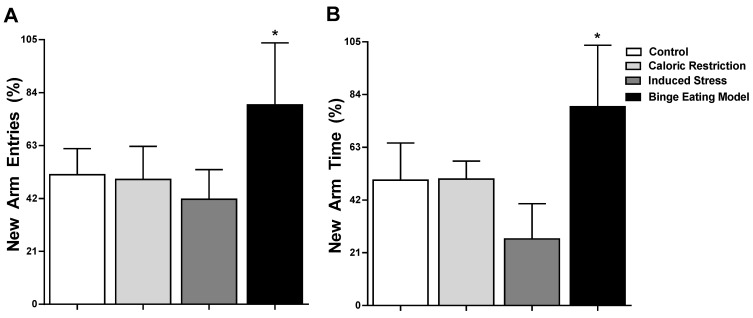
Perseverative-like behavior. (**A**) Number of entries to the new arm (in percentage) of the four groups of rats during the first 2 min of the test. (**B**) Time spent in the new arm (in percentage) of the four groups of rats during the first 2 min of the test. * *p* < 0.05.

**Table 1 nutrients-16-01275-t001:** Stress induction in each cycle of caloric restriction–refeeding.

	Adaptation	Cycle 1			Cycle 2			Cycle 3		
Group	3 Days	Days 1–4	Days 5–6	Days 7–8	Days 9–12	Days 13–14	Days 15–16	Days 17–20	Days 21–24	Day 25
Control	Ad lib food	Ad lib food	Ad lib food + HPF (2 h)	Ad lib food	Ad lib food	Ad lib food + HPF (2 h)	Ad lib food	Ad lib food	Ad lib food	No stress+ ad lib food + HPF (2 h)
Caloric Restriction	Ad lib food	Food restriction	Ad lib food + HPF (2 h)	Ad lib food	Food restriction	Ad lib food + HPF (2 h)	Ad lib food	Food restriction	Ad lib food	No stress + ad lib food + HPF (2 h)
Induced Stress	Ad lib food	Ad lib food	Stress + ad lib food + HPF (2 h)	Ad lib food	Ad lib food	Stress + ad lib food + HPF (2 h)	Ad lib food	Ad lib food	Ad lib food	Stress + ad lib food + HPF (2 h)
Binge Eating Model	Ad lib food	Food restriction	Stress + Ad lib food + HPF (2 h)	Ad lib food	Food restriction	Stress + ad lib food + HPF (2 h)	Ad lib food	Food restriction	Ad lib food	Stress + ad lib food + HPF (2 h)

## Data Availability

The data presented in this study are available on request from the corresponding author due to privacy reasons.

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
