# Peer review of "Increased Depressive-like, Anxiety-like, and Perseverative-like Behavior in Binge Eating Model in Juvenile Rats"

_nutrients, 2024, doi:10.3390/nu16091275_

Round 1
Reviewer 1 Report (New Reviewer)
Comments and Suggestions for Authors
The present manuscript examined the effect of binge-like eating on depressive-, anxiety-, and perseverative-like behaviors in male rats. The authors concluded that calorie restriction-refeeding along with stress may trigger depressive-, anxiety-, and perseverative-like behaviors.
While the study is based on straight-forward experiments and focus on an important topic, several concerns listed below must be addressed before it is suitable for publication.
Abstract does not define groups and without this information it is confusing to comprehend the outcomes (especially line 19-22). In the end, the authors mention groups, but which groups?
Line 64 – rewrite to clarity.
Introduction – Last three paragraphs: The authors here try to present the rationale and justification for the current study. However, a coherent and clear picture of study novelty is missing.
Line 94 states that 30 days old rats were used. If experimental procedure started during adolescent period, it must be clearly mentioned in the title and abstract. It is mentioned in the discussion though.
Line 100 states there were n=8/cage. A total of 32 animals were used. Were the animals group housed then? How was food assessment carried out?
How was the group assignment done to ensure that there was no baseline food, water and body weight difference?
Line 128: It appears that experimental manipulations were conducted during the light cycle. Please explain the rationale. Would authors expect different results if same study was conducted in the subjective dark cycle (active time) of the rodents.
Figure 2: The authors need to figure out a better way to present these data. The vertical lines for the cycles and palatable food intake look identical in the graphs which is confusing. Also, why there are no error bars? Enlarge the symbols for clarity.
The authors designated group 4 as binge eating model. Based on their experimental procedure, there were at least three different sessions where palatable food was provided. Yet, excessive (binge-like) intake was observed only during the second session which did not persist in the next session. On the other hand, several studies providing time-restricted feeding of a palatable diet show repeated bouts of binge-like eating which sustain several weeks. This basically undermines the construct validity of the study.
Figure 2C: When was body weight taken? Clearly mention these details in section 2.2. Several studies providing time-restricted feeding of a palatable diet show no difference in body weight. Could this be the stress+calorie restriction or the time of the diet presentation (light vs dark)?
Figure 3: The authors concluded that calorie restriction-refeeding along with stress may trigger depressive-like behavior. But there was no difference between stress and binge eating group of rats. Therefore, their conclusion is not valid.
Figure 4B: While corresponding results discussion (Line 360-366) report statistical differences, Figure 4B does not show any symbols. Not sure what is correct.
Why Y-maze data were not shown? The corresponding result section stated significant differences, but the Figure was missing.
Few typographical errors.
Comments on the Quality of English Language
Overall Ok. Few typographical errors.
Author Response
Review #1
The present manuscript examined the effect of binge-like eating on depressive-, anxiety-, and perseverative-like behaviors in male rats. The authors concluded that calorie restriction-refeeding along with stress may trigger depressive-, anxiety-, and perseverative-like behaviors.
While the study is based on straight-forward experiments and focus on an important topic, several concerns listed below must be addressed before it is suitable for publication.
Abstract does not define groups and without this information it is confusing to comprehend the outcomes (especially line 19-22). In the end, the authors mention groups, but which groups?
Response: Thank you to reviewed #1 for his comment. We have rewrite the introduction, methods and results of the Abstract section.
Change in the manuscript.
The aim of the present study was to evaluate depressive-like, anxiety-like and perseverative-like behaviors in a binge eating model. Wistar rats, using the binge eating model, was compared to the caloric restriction, induced stress and control groups. Rats of induced stress group presented binge-like behaviors in standard food intake in the second cycle of the experiment when compared to the caloric restriction group and the binge eating model group. Depressive-like behavior was observed in binge eating model group with longer immobility time (p < 0.001) and less time swim (p < 0.001) in comparison to the control group
Line 64 – rewrite to clarity.
Response: We agree. We rewrite the sentence for clarity.
Change in the manuscript. Line 64. Up to now, all the animal models in binge-like eating are type isomorphic models.
Introduction – Last three paragraphs: The authors here try to present the rationale and justification for the current study. However, a coherent and clear picture of study novelty is missing.
Response: We deeply appreciate the reviewer´s comment, and have rewrite the last three paragraphs in the revised manuscript.
Change in the manuscript. Page 2, Lines 72 – 84.
Some studies using animal models have suggesting that caloric restriction and stress promote binge-like eating [13], caloric restriction increases sensitivity to hyperphagic effect [14] and increased expression of sterol regulatory element binding protein in the hypothalamus [15], at the same time, binge eating produces dopamine and μ-opioid receptor dysregulation in the brain [16]. Moreover, access to highly palatable foods (HPF) increases anxiety-like behavior [17, 18] and food addiction [19].
There is limited evidence in animal models of binge-like eating that simultaneously assess depressive-like, anxiety-like and perseverative-like behaviors [20]. Therefore, the aim of this study was to assess depressive-like, anxiety-like and perseverative-like behaviors in a Wistar rats in binge eating, caloric restriction, induce stress and control group. We used the forced swimming test to assess depressive-like behaviors [21], the defensive burying test to assess anxiety-like behaviors [22], and the Y-maze test to assess perseverative-like behaviors [23].
Line 94 states that 30 days old rats were used. If experimental procedure started during adolescent period, it must be clearly mentioned in the title and abstract. It is mentioned in the discussion though.
Response: We agree. We finalized our experimental procedure to 55 days of life of the rats. Then we worked with juvenile rats. Then, we change the title and the manuscript for showed this in the present revised version.
Change in the manuscript.
Title: Page 1, Line 2-3. Increased depressive-like, anxiety-like, and perseverative-like behavior in binge eating model in juvenile rats.
Introduction: Page 2, Line 81 Therefore, the aim of this study was to assess depressive-like, anxiety-like and perseverative-like behaviors in a juvenile Wistar rats in binge eating…
Discussion: Page 11, Line 395-400. The aim of the present study was to evaluate anxiety-like, depression-like and perseverative-like behaviors in a juvenile binge eating model.
Conclusion: Page 13, Line 497-499. These results suggest that caloric restriction-refeeding coupled with repeated induction of stress, may be useful in the analysis of behavioral alterations such as depressive-like, anxiety-like and perseverative-like in juvenile rats with binge-like eating
Line 100 states there were n=8/cage. A total of 32 animals were used. Were the animals group housed then? How was food assessment carried out?
Response: Correct, there were 32 animals in total, 8 per box. The foods were measured and weighed daily. This is explained in the methodology section.
How was the group assignment done to ensure that there was no baseline food, water and body weight difference?
Response: All groups had an adaptation period for 3 days, where they were given food at libitum.
Line 128: It appears that experimental manipulations were conducted during the light cycle. Please explain the rationale. Would authors expect different results if same study was conducted in the subjective dark cycle (active time) of the rodents.
Response. The change from light cycle was carried out from his birth. This change is made to be able to evaluate the behaviors of nocturnal animals. If this change is not made, behavioral evaluations would have to be at night, when they are active.
Figure 2: The authors need to figure out a better way to present these data. The vertical lines for the cycles and palatable food intake look identical in the graphs which is confusing. Also, why there are no error bars? Enlarge the symbols for clarity.
Response: We appreciate the reviewer's suggestion. So as not to confuse readers. The authors decided not to make changes. We hope the reviewer can understand us.
The authors designated group 4 as binge eating model. Based on their experimental procedure, there were at least three different sessions where palatable food was provided. Yet, excessive (binge-like) intake was observed only during the second session which did not persist in the next session. On the other hand, several studies providing time-restricted feeding of a palatable diet show repeated bouts of binge-like eating which sustain several weeks. This basically undermines the construct validity of the study.
Response: Eating disorders are heterogeneous disorders with unresolved etiologies. One of the most accepted hypotheses is the constant exposure to stress generated by constantly restricting food (being on a diet). We tried to reproduce these binge-restriction cycles, which had previously been covered in models, Boggiano y del modelo Cifani (Boggiano & Chandler, 2006; Cifani et al., 2009). However, we need more restriction cycles to better evaluate binge eating. Let's hope to be able to do this in the future.
Figure 2C: When was body weight taken? Clearly mention these details in section 2.2. Several studies providing time-restricted feeding of a palatable diet show no difference in body weight. Could this be the stress+calorie restriction or the time of the diet presentation (light vs dark)?
Response: We believe weight gain is due to cycles. Stress +calorie restriction. Although the model needs to be evaluated with more cycles
Figure 3: The authors concluded that calorie restriction-refeeding along with stress may trigger depressive-like behavior. But there was no difference between stress and binge eating group of rats. Therefore, their conclusion is not valid.
Response: Figure 3 shows a longer swimming time in the model with respect to the control and less immobility of the model with respect to the control, the decrease in immobility shows behavioral hopelessness in the model (Stress +calorie restriction). It is important to mention that, individually, calorie restriction and stress do not show this type of behavior.
Figure 4B: While corresponding results discussion (Line 360-366) report statistical differences, Figure 4B does not show any symbols. Not sure what is correct.
Response: We agree. In this revised version, we checked the figure 4. We add symbols and add the color for each group.
Why Y-maze data were not shown? The corresponding result section stated significant differences, but the Figure was missing.
Response: We deeply appreciate the reviewer´s suggestion, and have add the figure 5. Now, we show the Y-maze data.
Few typographical errors.
Response: We apologize for the mistakes in the manuscript. We also carefully checked and corrected the entire manuscript for typographic errors.
Reviewer 2 Report (New Reviewer)
Comments and Suggestions for Authors
1. Change "Rats with repeated stress induction presented binge-like behaviors" to "Rats subjected to repeated stress induction exhibited binge-eating-like behaviors."
2. The structure of the manuscript is logical, with a clear progression from introduction, methodology, results, and discussion. However, the transition between sections could be smoother, and some sections could benefit from further subdivision to enhance readability and organization.
3. Introduce subheadings within the methodology and results sections to guide the reader more effectively through the study's design and findings.
4. This study addresses an important gap in the understanding of binge eating behaviors in rats using a novel combination of stress and caloric restriction to simulate these conditions. However, the study highlights lacks information how this approach differs from and improves upon the existing models.
5. Emphasize the novel aspects of your methodology and experimental design more explicitly in the Introduction and Discussion. Compare your approach with existing models to highlight their innovative aspects.
6. Explicitly state how your model of inducing binge eating behaviors represents an advancement over the Cifani model in the introduction.
7. Strengthen the discussion by more deeply integrating your findings with existing research, identifying areas where your results support, contradict, or extend previous studies.
8. Discuss the limitations more thoroughly, including potential biases, the generalizability of the findings to other populations (e.g., humans), and any factors that might limit the interpretation of the results.
Comments on the Quality of English Languagequite good, needs minor changes
Author Response
Review #2
- Change "Rats with repeated stress induction presented binge-like behaviors" to "Rats subjected to repeated stress induction exhibited binge-eating-like behaviors."
Response: Done.
Change in the manuscript. Page 13. Line 502-504.
- The structure of the manuscript is logical, with a clear progression from introduction, methodology, results, and discussion. However, the transition between sections could be smoother, and some sections could benefit from further subdivision to enhance readability and organization.
Response: We thank reviewer #2 for this insightful comment. In this revised manuscript, we performed change in the large of the manuscript. Abstract, introduction and conclusion sections.
Page 1. Lines 16-23. Page 2, Line, 72-84. And Page 11, Lines 404-409.
- Introduce subheadings within the methodology and results sections to guide the reader more effectively through the study's design and findings.
Response: We appreciate this valuable comment. In this revised manuscript, we have many changes to long of the manuscript. We hope that it could be helpful for the reader, and made this manuscript easier reader.
- This study addresses an important gap in the understanding of binge eating behaviors in rats using a novel combination of stress and caloric restriction to simulate these conditions. However, the study highlights lacks information how this approach differs from and improves upon the existing models.
Response: We agree. We have made changes in the discussion section to make a strength of the present study and show the information how our approach differs of the existent models.
Response: Our animal model is an adaptation of characteristics of the Boggiano model and the Cifani model (Boggiano & Chandler, 2006; Cifani et al., 2009). The cycles are combined (Stress +calorie restriction) highlighting that with this combination, compulsion in eating and weight (binge eating) is observed. In addition, 5 tests are measured to determine behavioral hopelessness, anxiety and compulsion. tests that had not been carried out on a similar model. The same thing that is suggested can be improved by increasing the number of cycles to work on a cleaner model of binge eating disorder.
- Emphasize the novel aspects of your methodology and experimental design more explicitly in the Introduction and Discussion. Compare your approach with existing models to highlight their innovative aspects.
Response: Thank reviewer #2 for this valuable comment. According to your suggestion, we have change in the introduction and discussion section in this paper.
Change in the manuscript.
Page 2, Lines 72 to 84. Some studies using animal models have suggesting that caloric restriction and stress promote binge-like eating [13], caloric restriction increases sensitivity to hyperphagic effect [14] and increased expression of sterol regulatory element binding protein in the hypothalamus [15], at the same time, binge eating produces dopamine and μ-opioid receptor dysregulation in the brain [16]. Moreover, access to highly palatable foods (HPF) increases anxiety-like behavior [17, 18] and food addiction [19].
There is limited evidence in animal models of binge-like eating that simultaneously assess depressive-like, anxiety-like and perseverative-like behaviors [20]. Therefore, the aim of this study was to assess depressive-like, anxiety-like and perseverative-like behaviors in a juvenile Wistar rats in binge eating, caloric restriction, induce stress and control group. We used the forced swimming test to assess depressive-like behaviors [21], the defensive burying test to assess anxiety-like behaviors [22], and the Y-maze test to assess perseverative-like behaviors [23].
Page 11, Line 404-409. The aim of the present study was to evaluate anxiety-like, depression-like and perseverative-like behaviors in a juvenile binge eating model. There are few studies that have evaluated the presence of behavioral alterations such as depression, anxiety and compulsion in the same group of rats in a binge eating model [40, 41]. Our study showed that calorie restriction-refeeding along with stress, may lead to anxiety-like, depression-like and perseverative-like behaviors are present in rats in a binge eating model.
Page 12, Line 440-444. We considered that this type of stress is similar to those on restrictive diets, where people who want to lose weight restrict palatable foods, with the desire to lose weight, but this causes them stress. Provoking that when they finish the diet, they compulsively eat these palatable foods again.
- Explicitly state how your model of inducing binge eating behaviors represents an advancement over the Cifani model in the introduction.
Response: Thank you, this suggestion, we add the sentence “we, used a combined of stress + calorie restriction and a combination, compulsion in eating and weight. We considered as and advancement over the previously models.”. We considered that this sentence could resume the advance of the present model over the Cifani model.
Page 2, Line 85-87.
- Strengthen the discussion by more deeply integrating your findings with existing research, identifying areas where your results support, contradict, or extend previous studies.
Response. Thank you, this suggestion was carried out in the writing
- Discuss the limitations more thoroughly, including potential biases, the generalizability of the findings to other populations (e.g., humans), and any factors that might limit the interpretation of the results.
Response. Emphasis was placed on the limitations. thanks for the observation.
Comments on the Quality of English Language
quite good, needs minor changes
Response: We appreciate the valuable comments. Since, we are not a native English speaker, this manuscript was proofread by an English speaking professional with science background.
Reviewer 3 Report (New Reviewer)
Comments and Suggestions for Authors
This is an interesting paper on the topic.
The authors are kindly invited to revisit the following points:
Lines 35-43: it seems that the authors have not cited the proper references in the respective text. Please revisit and potentially include additional references to support your statements (particularly in lines 39-42)
Lines 43-44: kindly note that the references included in this statement refer to specific populations, perhaps the authors would like to consider including research on a broader spectrum of participants in order to generalize the association.
Lines 44-45: Please revisit the reference for suitability in supporting the statement.
Line 47: "Binge eating disorder is the most common eating disorder." Kindly include the source of information regarding this statement.
Line 48: "Studies in adolescent population" The authors have included only one relevant reference following this statement. Kindly revisit and rephrase accordingly. If it is only one study please consider adding information on the population characteristics i.e. country, health status etc. If the authors wish to keep the statement as it is, further insights must be included (additional studies on different populations, countries, time periods, etc).
Line 56: "Also, binge eating disorder interferes with decision making." Please include relevant references to support your statement.
Line 60: "To improve our understanding of binge eating, a proper characterization of animal models is necessary". How is this supported? Also, there is a big gab in the conseptual sequence here - From suggesting that bidge eating is relevant for all adolecents to the need of animal model characterization - Kindly revisit and rephrase so that the reader will be able to follow your line of thinking and presenting this work.
Line 55: kindly revisit the typo.
Lines 64-65: Statement is not clear and it lacks reference. Please revisit and rephrase.
In overall the Introduction must be revsitied as it does not provide the reader with vital information on the research gab that this work aims to adress.
Lines 97 - 100: Kindly revisit the description of methods. Perhaps the authors would like to consider a cleared description and revisit the use of english.
Lines 108-109: Perhaps it would be helpful for the reader to understand "what" and "why" the authors needed to change in the original protocols. Also having a clear understanding of the scope of those chages will allow future researhers to use this work as a reference point for thei own work. Please consider elaborating and presenting the protocol as clearly as possible (just like tha authors already do further in the manuscript in the description of the tests).
Figures 2 and 3 are very difficult to read. Please consider vertical presentation to ensure larger images.
Lines 266, 274, 313 and 321: Perhaps the authors would like to elaborate on the difference between Tukey’s post hoc test and the Post-test analysis. It is not clear what the second analysis refers to (is it a t-test done here?)
Lines 356-357: "The control groups (34.86 ± 12.68 s) and eating model showed" Kindly revisit if this phrase is correct.
Figure 4 is missing information on the caption of the groups/colors.
Regarding the results, it may be beneficial for the reader to have them presented in a table rather than in the text. Kindly consider this option of having a table of the means and SDs and next to them the p values indicating the significantly different variables identified. This is just a suggestion and the authors should feel free to discard it.
Line 415: "This, according with Spear (2000), He reported that, adolescents" Please revisit and rephrase.
Line 428: "as is known, eating disorders occur more frequently in women, however, binge eating disorder is more common in men." How is this statement supported?
Lines 456-458: "Our results suggest the presence of depressive-like behavior by longer immobility time and shorter time spent swimming, in rats with only stress induction and in rats of the binge eating model group, compared to the control group." Is this statement accurate? Kindly revisit.
Line 486: Revisit for typo.
Comments on the Quality of English LanguageMinor check is requred
Author Response
Reviewer #3
Comments and Suggestions for Authors
This is an interesting paper on the topic.
Response: Thank you
The authors are kindly invited to revisit the following points:
Lines 35-43: it seems that the authors have not cited the proper references in the respective text. Please revisit and potentially include additional references to support your statements (particularly in lines 39-42)
Response: Done.
Lines 43-44: kindly note that the references included in this statement refer to specific populations, perhaps the authors would like to consider including research on a broader spectrum of participants in order to generalize the association.
Response: Done
Lines 44-45: Please revisit the reference for suitability in supporting the statement.
Response. Done
Line 47: "Binge eating disorder is the most common eating disorder." Kindly include the source of information regarding this statement.
Response: Done
Line 48: "Studies in adolescent population" The authors have included only one relevant reference following this statement. Kindly revisit and rephrase accordingly. If it is only one study please consider adding information on the population characteristics i.e. country, health status etc. If the authors wish to keep the statement as it is, further insights must be included (additional studies on different populations, countries, time periods, etc).
Response: We agree. We add the references that we used in the paragraph.
Line 56: "Also, binge eating disorder interferes with decision making." Please include relevant references to support your statement.
Response: Done.
Line 60: "To improve our understanding of binge eating, a proper characterization of animal models is necessary". How is this supported? Also, there is a big gab in the conseptual sequence here - From suggesting that bidge eating is relevant for all adolecents to the need of animal model characterization - Kindly revisit and rephrase so that the reader will be able to follow your line of thinking and presenting this work.
Response: We thank reviewer. We have change and add the sentence “However, are necessary model that explain this behavior in binge eating. To improve our understanding of animal models for binge eating, a proper characterization of animal models is necessary”, changes in the manuscript. Page 2, Line 59-61.
Line 55: kindly revisit the typo.
Response: Done
Lines 64-65: Statement is not clear and it lacks reference. Please revisit and rephrase.
In overall the Introduction must be revsitied as it does not provide the reader with vital information on the research gab that this work aims to adress.
Response: We appreciate this valuable comment. In this revised manuscript, we have many changes to long of the manuscript, in special in the introduction section. We hope that it could be helpful for the reader, and made this manuscript easier reader.
Lines 97 - 100: Kindly revisit the description of methods. Perhaps the authors would like to consider a cleared description and revisit the use of English.
Response: Done. We have changed the sentence. Page 3, Lines 101-103. Bioethics was considered in the use of experimental animals [27], as well as criteria for the handling and use of experimental animals by the internal committee of UJAT-DACS (Comité Interno para el Cuidado y Uso de los Animales de Laboratorio – CICUAL).
Lines 108-109: Perhaps it would be helpful for the reader to understand "what" and "why" the authors needed to change in the original protocols. Also having a clear understanding of the scope of those chages will allow future researhers to use this work as a reference point for thei own work. Please consider elaborating and presenting the protocol as clearly as possible (just like tha authors already do further in the manuscript in the description of the tests).
Response: Thank reviewer #3 for this valuable comment. According to your suggestion, we have change in the introduction and discussion section in this paper.
Change in the manuscript.
Page 2, Lines 72 to 84. Some studies using animal models have suggesting that caloric restriction and stress promote binge-like eating [13], caloric restriction increases sensitivity to hyperphagic effect [14] and increased expression of sterol regulatory element binding protein in the hypothalamus [15], at the same time, binge eating produces dopamine and μ-opioid receptor dysregulation in the brain [16]. Moreover, access to highly palatable foods (HPF) increases anxiety-like behavior [17, 18] and food addiction [19].
There is limited evidence in animal models of binge-like eating that simultaneously assess depressive-like, anxiety-like and perseverative-like behaviors [20]. Therefore, the aim of this study was to assess depressive-like, anxiety-like and perseverative-like behaviors in a juvenile Wistar rats in binge eating, caloric restriction, induce stress and control group. We used the forced swimming test to assess depressive-like behaviors [21], the defensive burying test to assess anxiety-like behaviors [22], and the Y-maze test to assess perseverative-like behaviors [23].
Page 11, Line 404-409. The aim of the present study was to evaluate anxiety-like, depression-like and perseverative-like behaviors in a juvenile binge eating model. There are few studies that have evaluated the presence of behavioral alterations such as depression, anxiety and compulsion in the same group of rats in a binge eating model [40, 41]. Our study showed that calorie restriction-refeeding along with stress, may lead to anxiety-like, depression-like and perseverative-like behaviors are present in rats in a binge eating model.
Page 12, Line 440-444. We considered that this type of stress is similar to those on restrictive diets, where people who want to lose weight restrict palatable foods, with the desire to lose weight, but this causes them stress. Provoking that when they finish the diet, they compulsively eat these palatable foods again.
Figures 2 and 3 are very difficult to read. Please consider vertical presentation to ensure larger images.
Response: In this revised version, we checked the figure 2 and 3. Also, we revised the figure 4 and included a new figure the 5.
Figure 3 shows a longer swimming time in the model with respect to the control and less immobility of the model with respect to the control, the decrease in immobility shows behavioral hopelessness in the model (Stress +calorie restriction). It is important to mention that, individually, calorie restriction and stress do not show this type of behavior.
Lines 266, 274, 313 and 321: Perhaps the authors would like to elaborate on the difference between Tukey’s post hoc test and the Post-test analysis. It is not clear what the second analysis refers to (is it a t-test done here?)
Response: Thank reviewer #3 for this comment. In consequence we have rewrite the statistical analysis section. This for a clarify the use and the ANOVA and Tukey´s post hoc test.
Change in the manuscript. Page 7, lines 250-25. We used ANOVA test to assess the effect of caloric restriction-refeeding cycles between groups. We used one-way ANOVA to assess the effect of caloric restriction-refeeding cycles on the test that we used for measured of depressive like, anxiety-like and perseverative-like behavior. Tukey’s post hoc test was used to compare the means differences values of the groups. The level of statistical significance was set at p < 0.05. Data are presented as means ± standard error of mean. The data were analyzed by GraphPad Prism 6.1 software (GraphPad Software, Boston, MA, USA).
Lines 356-357: "The control groups (34.86 ± 12.68 s) and eating model showed" revisit if this phrase is correct.
Response: Corrected.
Figure 4 is missing information on the caption of the groups/colors.
Response: We agree. In this revised version, we checked the figure 4. We add symbols and add the color for each group.
Regarding the results, it may be beneficial for the reader to have them presented in a table rather than in the text. Kindly consider this option of having a table of the means and SDs and next to them the p values indicating the significantly different variables identified. This is just a suggestion and the authors should feel free to discard it.
Response: We agree with reviewer "3. In future work, we want to present the results in tables rather than in text.
Line 415: "This, according with Spear (2000), He reported that, adolescents" Please revisit and rephrase.
Response: Thank you, this suggestion. For make more easy reader we deleted this sentence.
Line 428: "as is known, eating disorders occur more frequently in women, however, binge eating disorder is more common in men." How is this statement supported?
Response: Done
Lines 456-458: "Our results suggest the presence of depressive-like behavior by longer immobility time and shorter time spent swimming, in rats with only stress induction and in rats of the binge eating model group, compared to the control group." Is this statement accurate? Kindly revisit.
Response: We agree.
Line 486: Revisit for typo.
Response: Checked
Comments on the Quality of English Language
Minor check is requred
Round 2
Reviewer 3 Report (New Reviewer)
Comments and Suggestions for Authors
Thank you to the authors for revisiting the manuscript which has been significantly improved.
Kindly consider the following points.
Lines 59-60: "However, are necessary model that explain this behavior in binge eating." This doesn't read correctly. Please rephrase.
Line 249: Kindly indicate if the assumption of normality of data distribution is evaluated in the samples before ANOVA is used and include in the results.
Line 448: Finally, re recognized some limitation in our study. Revisit for typo.
Comments on the Quality of English Language
Minor revisions are needed
Author Response
Thank you to the authors for revisiting the manuscript which has been significantly improved.
Kindly consider the following points.
Lines 59-60: "However, are necessary model that explain this behavior in binge eating." This doesn't read correctly. Please rephrase.
Response: We thank reviewer #2 fir this comment. We have rephrased the sentence
Change in the manuscript. Page 2, Line 59-60. Therefore, models are necessary to explain this behavior in binge eating.
Line 249: Kindly indicate if the assumption of normality of data distribution is evaluated in the samples before ANOVA is used and include in the results.
Response: We appreciate this valuable comment. We have evaluated the distribution and add the sentence in the statistics analysis subsection.
Change in the manuscript. Page 7, Lines 250-251. To determine the normality of the data, we performed a Shapiro-Wilk test. The variables had a normal distribution (p-value <0.05).
Line 448: Finally, re recognized some limitation in our study. Revisit for typo.
Response: Done
Page 12, Line 449 to 452. It might be that more cycles of caloric restriction-refeeding favors binge-like eating in highly palatable food intake, as indicated by the findings of Hagan and Moss, where rats manifested binge-like eating up to 12 weeks of refeeding-restriction were needed [53].
This manuscript is a resubmission of an earlier submission. The following is a list of the peer review reports and author responses from that submission.
Round 1
Reviewer 1 Report
Comments and Suggestions for Authors
Review of "Increased depressive-like, anxiety-like, and perseverative-like behavior in binge eating model in rats" by Alma Delia Genis Mendoza et al.
The topic of binge eating behavior may be interested by many researchers, thus this report may have a decent impact on the field. The study is scientifically sound and the data looks solid. I have a bit concern of the potential novelty, as many conclusions of this manuscript have been reported already. The authors discussed the similarities and differences between their study and previous articles, which is good. Expanding on this texts, the authors should not only point out the observations but also explain (or try to explain) why these differences occurred. In addition, compare the rat model with patients studies is also needed. Adding this information would increase the impact of current study to the research field.
Author Response
Review #1
Review of "Increased depressive-like, anxiety-like, and perseverative-like behavior in binge eating model in rats" by Alma Delia Genis Mendoza et al.
The topic of binge eating behavior may be interested by many researchers, thus this report may have a decent impact on the field. The study is scientifically sound and the data looks solid. I have a bit concern of the potential novelty, as many conclusions of this manuscript have been reported already. The authors discussed the similarities and differences between their study and previous articles, which is good. Expanding on this texts, the authors should not only point out the observations but also explain (or try to explain) why these differences occurred. In addition, compare the rat model with patients studies is also needed. Adding this information would increase the impact of current study to the research field.
Response: Thank you for your valuable comments. In this revised version, we included try to explain the comparison the rat model with patients studies.
Change in the manuscript. Discussion section.
Page 11, Line 49 to 51. In this work we only evaluated male Wistar rats, a little to explore the behavior without other variables, however it is necessary to evaluate these behaviors in female Wistar rats, since, as is known, eating disorders occur more frequently in women, without However, binge eating disorder is more common in men (Martinez Magaña 2022).
Page 12, Line 61 to 65. We believe that this type of stress is similar to those on restrictive diets, where people who want to lose weight restrict palatable foods, with the desire to lose weight, but this causes them stress. Provoking that when they finish the diet, they compulsively eat these palatable foods again.
Page 12, lines 84 to 86. As we have already pointed out, eating disorders are complex with several associated comorbidities, such as depression (Ramos-Ruiz, 2021)
Reviewer 2 Report
Comments and Suggestions for Authors
The manuscript provides insights into the effects of caloric restriction-refeeding cycles and repeated stress induction on depressive-like, anxiety-like, and perseverative-like behaviors in a binge eating model using male Wistar rats.
1. (line 43, line 54) You mentioned it is suggested that self-imposed dieting coupled with overeating, precede binge eating behaviors in humans. And binge eating disorder affects both male and female adolescents. However, this study is conducted only on male Wistar rats, which may limit the generalizability of the findings to other strains or sexes of rats, as well as to humans. Please explain why you choose male Wistar rats with thirty days old as animal model. What is the reason of selection of thirty days-old rats? Is it comparable to humans’ adolescents?
2. (Line 275) You stated the statistical significance as p ≤0.05. Usually, p-value<0.05 used to be set for statistical significance. What is the reason for this set?
3. (Line 357, line 358, line 364, line 381) you described p<0.05 for the results. Then, were they not statistically significant?
4. (Line 370) Figure legend of figure 3 showed p<0.05 was set as statistical significant. Please correct or explain.
5. (Line 274 and Line 306) you pointed out the use of Tukey’s post hoc test. Why do you use it for the analysis? Please mention any brief description in the statistical analysis section and what is the meaning for the analysis for this study? There is no any explanation for “F value”. Please explain this as well.
6. (line 398) What is Gray’s scale? I cannot see the explanation of what it is throughout the manuscript.
7. (line 420) I cannot understand what the aim of this study is in this phrase. You mentioned your previous study results [40, 41].
8. (line 437) Then, what is your difference of this study results from other Boggianop and Chandler and Cifani et al., except for the animal sex? Please make sure clearly describe your results.
Author Response
Reviewer 2
The manuscript provides insights into the effects of caloric restriction-refeeding cycles and repeated stress induction on depressive-like, anxiety-like, and perseverative-like behaviors in a binge eating model using male Wistar rats.
(line 43, line 54) You mentioned it is suggested that self-imposed dieting coupled with overeating, precede binge eating behaviors in humans. And binge eating disorder affects both male and female adolescents. However, this study is conducted only on male Wistar rats, which may limit the generalizability of the findings to other strains or sexes of rats, as well as to humans. Please explain why you choose male Wistar rats with thirty days old as animal model. What is the reason of selection of thirty days-old rats? Is it comparable to humans’ adolescents?
Response: We deeply appreciate the reviewer´s comment. And have added the discussion about the use of male rats. And, we recognized this as a limitation of the present study.
Change in the manuscript.
Page 13, Lines 501 to 520.
Finally, re recognized some limitation in our study. We used only male rats. Eating disorders occur more in women than in men, although recently an increase has been observed in men. Working with male rats facilitated behavioral readings, without taking into account hormonal variations. However, the next challenge is also working with female rats. Second. We measured in two face 30 or 90 days. The eating disorders occurred at the beginning of adolescence, however, it is increasingly occurring at younger ages. Studying 30-day-old animals helped us see similarities in adolescence. Our group has studied various ages of 45 and 90 days, with similarities in adolescent and adult behavior, where it is possible to observe characteristics associated with psychiatric disorders, especially with schizophrenia.
2(Line 275) You stated the statistical significance as p≤0.05. Usually, p-value<0.05 used to be set for statistical significance. What is the reason for this set?
Response: We set the significance in p=0.05.
- (Line 357, line 358, line 364, line 381) you described p<0.05 for the results. Then, were they not statistically significant?
Response: We now set the significance in p=0.05. Then the results are statistically significant.
4.(Line 370) Figure legend of figure 3 showed p<0.05 was set as statistical significant. Please correct or explain.
Response: Corrected.
- (Line 274 and Line 306) you pointed out the use of Tukey’s post hoc test. Why do you use it for the analysis? Please mention any brief description in the statistical analysis section and what is the meaning for the analysis for this study? There is no any explanation for “F value”. Please explain this as well.
Response: Thank you for your comment. In this revised version, we rewrite the statistical analysis subsection, this for clarify the use of ANOVA and the p-value.
Change in the manuscript
Page 7, Lines 274-281
We used ANOVA test to assess the effecto of caloric restriction-refeeding cycles between groups. We used two-way ANOVA to assess the effect of caloric restriction-refeeding cycles on the test that we used for measured of depressive like, anxiety-like and persevarative-like behavior. Tukey's post hoc test was used to compare the means values of the groups. The level of statistical significance was set at p = 0.05. Data are presented as means, standard error of mean, standard deviations or mean differences. The graphical software used was GraphPad Prism
- (line 398) What is Gray’s scale? I cannot see the explanation of what it is throughout the manuscript.
Response: We apologize for the mistakes in the manuscript. We deleted this in this revised version.
- (line 420) I cannot understand what the aim of this study is in this phrase. You mentioned your previous study results [40, 41].
Response: We agree. This was deleted and rewrite in the paragraph
Change in the manuscript. Page 11, Lines 430.
- (line 437) Then, what is your difference of this study results from other Boggianop and Chandler and Cifani et al., except for the animal sex? Please make sure clearly describe your results.
Response: We deeply appreciate the reviewer’s comment. We want comment the following:
Eating disorders are complex disorders within the study of psychiatry, since they present associated comorbidities, such as anxiety, depression and obsessive-compulsive behavior towards food, causing alterations in weight perception and distorted eating behaviors. Despite their complexity, they are little studied. Therefore, an animal model helps to study these behaviors.
One of the main characteristics of binge eating disorder, in addition to eating large amounts of food in a short time, are obsessions and distorted compulsions regarding food and the way they view their body. Patients experience an uncontrollable desire to eat excessively. eating (compulsion) and the desire to continue eating (obsessions) despite being overweight. In animal models, it is difficult to measure these weight obsession behaviors that are typical of humans. In the project, compulsion-obsession-type behavior was measured using the Y-maze test, which had not been explored in these models.
Boggiano et al. 2006, he proposed a model with stress induced with electric shocks, food restriction of 66% and palatable food of Oreo cookies for 12 cycles, where he measured stress behavior. Cifaniy et al. 2009, proposes the most palatable foods made with Nutella cream, taking this mixture as a stress medium in cycles of 3 repetitions of exposure and restriction, in which anxiety and depression were modified. Our proposal was to combine some of the characteristics of both models, where it was possible to observe binge eating behavior and compulsion and/or obsession type behavior in the model. In addition to other behavioral tests such as forced swimming to observe depression-like behavior, open field, to observe locomotion, elevated cross-shaped maze, and defensive burying, to observe anxiety.
Although the model does not yield highly significant results, in its behaviors, it allows us to approach a binge eating model, with its main characteristics, such as anxiety, depression and obsession-compulsion. We must continue studying with females and perfect behavioral measurements, as well as molecular measurements that will surely help us explain the etiology of this complex disorder.
Round 2
Reviewer 2 Report
Comments and Suggestions for Authors
I have carefully reviewed the manuscript again. Despite my constructive feedback and concerns raised during the review process, the authors' responses lacked the expected sincerity and willingness to address the issues and recommendations. The authors' responses did not include all the necessary revisions or explanations to address the concerns raised in my initial review. In some cases, revised phrases were not provided one by one in the reply.
Author Response
Reviewer #2
I have carefully reviewed the manuscript again. Despite my constructive feedback and concerns raised during the review process, the authors' responses lacked the expected sincerity and willingness to address the issues and recommendations. The authors' responses did not include all the necessary revisions or explanations to address the concerns raised in my initial review. In some cases, revised phrases were not provided one by one in the reply.
Response:
We apologize the mistake in the response to comments made by Reviewer 2. In this review we consider all your comments and reviews of the first round. My colleagues and I believe this revision has been markedly improved. The answers to your remarks together with point to point clarifications are listed below; these include the changes asked and kindly suggested by the referees. We hope that reviewer #2 may consider accepting the manuscript.
Reviewer #2 first round.
The manuscript provides insights into the effects of caloric restriction-refeeding cycles and repeated stress induction on depressive-like, anxiety-like, and perseverative-like behaviors in a binge eating model using male Wistar rats.
(line 43, line 54) You mentioned it is suggested that self-imposed dieting coupled with overeating, precede binge eating behaviors in humans. And binge eating disorder affects both male and female adolescents. However, this study is conducted only on male Wistar rats, which may limit the generalizability of the findings to other strains or sexes of rats, as well as to humans. Please explain why you choose male Wistar rats with thirty days old as animal model. What is the reason of selection of thirty days-old rats? Is it comparable to humans’ adolescents?
Response: We deeply appreciate the reviewer´s comment. And have added the discussion about the use of male rats. And, we recognized this as a limitation of the present study. Change in the manuscript. Page 12, Lines 487 to 496. Finally, re recognized some limitation in our study. We used only male rats. Eating disorders occur more in women than in men, although recently an increase has been observed in men. Working with male rats facilitated behavioral readings, without taking into account hormonal variations. However, the next challenge is also working with female rats. Second. We measured in two face 30 or 90 days. The eating disorders occurred at the beginning of adolescence, however, it is increasingly occurring at younger ages. Studying 30-day-old animals helped us see similarities in adolescence. Our group has studied various ages of 45 and 90 days, with similarities in adolescent and adult behavior, where it is possible to observe characteristics associated with psychiatric disorders, especially with schizophrenia.
2(Line 275) You stated the statistical significance as p≤0.05. Usually, p-value<0.05 used to be set for statistical significance. What is the reason for this set?
Response: Now, we set the significance in p=0.05. Page 7, Line 257. The level of statistical significance was set at p = 0.05.
- (Line 357, line 358, line 364, line 381) you described p<0.05 for the results. Then, were they not statistically significant?
Response: We now set the significance in p=0.05. Then the results are statistically significant.
4.(Line 370) Figure legend of figure 3 showed p<0.05 was set as statistical significant. Please correct or explain.
Response: Corrected. Change in the manuscript. Page 11, Line 354. *p= 0.05, **p<0.01, ***p<0.001.
- (Line 274 and Line 306) you pointed out the use of Tukey’s post hoc test. Why do you use it for the analysis? Please mention any brief description in the statistical analysis section and what is the meaning for the analysis for this study? There is no any explanation for “F value”. Please explain this as well.
Response: Thank you for your comment. In this revised version, we rewrite the statistical analysis subsection, this for clarify the use of ANOVA and the p-value. Change in the manuscript. Page 7, Lines 253-259. We used ANOVA test to assess the effect of caloric restriction-refeeding cycles between groups. We used two-way ANOVA to assess the effect of caloric restriction-refeeding cycles on the test that we used for measured of depressive like, anxiety-like and perseverative-like behavior. Tukey’s post hoc test was used to compare the means differences values of the groups. The level of statistical significance was set at p = 0.05. Data are presented as means ± standard error of mean. The data were analyzed by GraphPad Prism 6.1 software (GraphPad Software, Boston, MA, USA).
- (line 398) What is Gray’s scale? I cannot see the explanation of what it is throughout the manuscript.
Response: We apologize for the mistakes in the manuscript. We deleted this in this revised version.
- (line 420) I cannot understand what the aim of this study is in this phrase. You mentioned your previous study results [40, 41].
Response: We agree. This was deleted and rewrite in the paragraph. Change in the manuscript. Page 11, Lines 402-408. The aim of the present study was to evaluate anxiety-like, depression-like and perseverative-like behaviors in a binge eating model. There are few studies that have evaluated the presence of behavioral alterations such as depression, anxiety and compulsion in the same group of rats in a binge eating model [40, 41]. Our study showed that calorie restriction-refeeding along with stress, may lead to anxiety-like, depression-like and perseverative-like behaviors are present in rats in a binge eating model.
- (line 437) Then, what is your difference of this study results from other Boggianop and Chandler and Cifani et al., except for the animal sex? Please make sure clearly describe your results.
Response: We deeply appreciate the reviewer’s comment. We want comment the following: Eating disorders are complex disorders within the study of psychiatry, since they present associated comorbidities, such as anxiety, depression and obsessive-compulsive behavior towards food, causing alterations in weight perception and distorted eating behaviors. Despite their complexity, they are little studied. Therefore, an animal model helps to study these behaviors. One of the main characteristics of binge eating disorder, in addition to eating large amounts of food in a short time, are obsessions and distorted compulsions regarding food and the way they view their body. Patients experience an uncontrollable desire to eat excessively. eating (compulsion) and the desire to continue eating (obsessions) despite being overweight. In animal models, it is difficult to measure these weight obsession behaviors that are typical of humans. In the project, compulsion-obsession-type behavior was measured using the Y-maze test, which had not been explored in these models. Boggiano et al. 2006, he proposed a model with stress induced with electric shocks, food restriction of 66% and palatable food of Oreo cookies for 12 cycles, where he measured stress behavior. Cifaniy et al. 2009, proposes the most palatable foods made with Nutella cream, taking this mixture as a stress medium in cycles of 3 repetitions of exposure and restriction, in which anxiety and depression were modified. Our proposal was to combine some of the characteristics of both models, where it was possible to observe binge eating behavior and compulsion and/or obsession type behavior in the model. In addition to other behavioral tests such as forced swimming to observe depression-like behavior, open field, to observe locomotion, elevated cross-shaped maze, and defensive burying, to observe anxiety. Although the model does not yield highly significant results, in its behaviors, it allows us to approach a binge eating model, with its main characteristics, such as anxiety, depression and obsession-compulsion. We must continue studying with females and perfect behavioral measurements, as well as molecular measurements that will surely help us explain the etiology of this complex disorder.